# Neuropathic pain in patients with osteoarthritis of the hip before and after total hip arthroplasty

**Yohei Yamabe, Masahiro Hasegawa** *, **Gai Kobayashi, Shine Tone, Yohei Naito, Akihiro Sudo**

Department of Orthopedic Surgery, Mie University Graduate School of Medicine, Mie, Japan

* masahase@clin.medic.mie-u.ac.jp

**Data Availability Statement:** All relevant data are within the manuscript and its Supporting Information files.

## Abstract

### Objectives

The pain associated with osteoarthritis (OA) was thought to be nociceptive; however, neuropathic pain is also observed. We investigated the relationship between hip OA and neuropathic pain using the PainDETECT questionnaire (PDQ).

### Methods

A total of 159 hips of 146 consecutive patients who underwent total hip arthroplasty (THA) with a diagnosis of OA were enrolled in this study. The prevalence of each pain phenotype was evaluated preoperatively and at 6 months postoperatively using the PDQ. Patient characteristics and numerical rating scale (NRS) scores were compared between a group with possible neuropathic pain (NP group) and a group with nociceptive pain (non-NP group).

### Results

Before THA, neuropathic, unclear, and nociceptive pain was observed in 18, 36, and 105 hips, respectively. The prevalence in the NP group was 54 hips, accounting for approximately one-third of all hips, which decreased significantly to seven hips after THA. A significantly higher NRS score was observed in the NP group, both before and after THA.

### Conclusion

Approximately one-third of the patients with hip OA had neuropathic pain. Therefore, neuropathic pain should be considered when treating patients with hip OA.

## Introduction

Persistent pain, a main symptom of osteoarthritis (OA), can cause joint dysfunction. The etiology of pain in OA is complex, and its exact pathophysiology is not completely understood.

**Funding:** The author(s) received no specific funding for this work.

**Competing interests:** he authors have declared that no competing interests exist.

Pain caused by inflammation is classified as nociceptive, whereas pain caused by nerve stimulation is classified as neuropathic pain. However, not all types of pain can be clearly categorized and many patients experience both types of pain [1]. For many years, pain associated with OA was thought to be nociceptive; however, recent reports indicate that neuropathic pain is involved as well [2]. Neuropathic pain is defined by the International Association for the Study of Pain as "pain arising as a direct consequence of a lesion or disease affecting the somatosensory system" [3]. Typical orthopedic conditions presenting with neuropathic pain include carpal tunnel syndrome and spinal disorders [4]. Neuropathic pain is characterized by tactile sensitivity and hyperalgesia, such as a burning sensation and numbness. Identifying patients with hip OA experiencing neuropathic pain is essential for providing patient education and additional treatment options beyond conventional pain treatment. Although a reliable and valid tool for assessing neuropathic pain symptoms in patients with OA is essential, few studies have explored the relationship between hip OA, neuropathic pain, and pain outcomes after total hip arthroplasty (THA). We hypothesized the existence of a neuropathic component in hip OA pain and that the occurrence of neuropathic pain in the hips of patients with OA might decrease after surgery. In this study, we evaluated neuropathic pain in patients with hip osteoarthritis before and after THA using the PainDETECT questionnaire (PDQ).

## Materials and methods

### Patients

We enrolled 146 consecutive patients with OA of the hip (159 hips; female, n = 126 [137 hips] male, n = 20 [22 hips]; mean age, 67 [28–88] years; mean body mass index (BMI), 25.3 [15.7–37.4] kg/m$^2$) who underwent THA from April 2020 to November 2022 prospectively. The surgical approaches included 95 superior, 35 posterior, 21 anterior, and 8 anterolateral approaches. Patients under 18 years of age, revision cases, those with dementia or psychiatric disorders, and those with missing questionnaire data were excluded. Patients whose preoperative physical and imaging findings suggested that their hip pain was due to spine-related disease were not indicated for THA. For the first two weeks after surgery, all patients took oral NSAIDs or acetaminophen according to our protocol. No patients used analgesics specifically for neuropathic pain.

### Pain phenotype

The PDQ was used to evaluate neuropathic hip pain. This questionnaire is a self-reporting pain-screening tool developed to assess the possibility that a patient's pain is neuropathic in usual practice, and can differentiate pain with 85% sensitivity and 80% specificity. It was originally developed for patients with chronic lower back pain [5] and has been used to treat various diseases [6]. The PDQ comprises nine questions regarding the intensity and quality of pain, and its scores are classified into pain course pattern, presence of radiating pain (maximum score of 3), and gradation of pain (maximum score of 35). The total score ranges from 0 to 38 points. Scores from 0 to 12 are classified as nociceptive pain, scores from 13 to 18 as unclear pain that includes elements of both nociceptive and neuropathic pain, and scores from 19 to 38 as neuropathic pain. In this study, unclear and neuropathic pain were grouped as possible neuropathic pain and evaluated in two groups: possible neuropathic pain (NP group: total 13–38 points) and nociceptive pain (non-NP group: total 0–12 points). The PDQ was administered preoperatively and 6 months postoperatively in all patients, and the prevalence of neuropathic pain was measured before and after THA by the surgeon. Sex, age, BMI, and numerical rating scale (NRS) scores on rest were compared between groups. The correlation between the preoperative PDQ score and the pre- and postoperative NRS scores and between

the preoperative PDQ score and BMI were also investigated. This study was approved by the institutional review board of Mie University Hospital (H2018-083), and all patients received written informed consent to participate. We conducted this study according to the principles of the Declaration of Helsinki.

## Statistical analyses

Mann-Whitney U test was used to compare pre- and postoperative NRS scores, age, and BMI between NP and non-NP groups. Chi-square test was used to compare gender and to compare the prevalence of possible neuropathic and nociceptive pain before and after THA. Spearman's correlation coefficient was used to determine the correlations between pre- and post-THA NRS scores and preoperative PDQ scores, and between BMI and preoperative PDQ scores. Fisher's test was used to assess the changes in pain type before and after THA. Statistical significance was set at $p < 0.05$. We use EZR (Saitama Medical Center, Jichi Medical University, Saitama, Japan), a graphical user interface for R (The R Foundation for Statistical Computing, Vienna, Austria) to analyze data [7].

## Results

Before THA, neuropathic, unclear, and nociceptive pain was present in 18 (11%), 36 (23%), and 105 (66%) hips, respectively. After THA, the number of hips with neuropathic and mixed pain decreased to 1 (1%) and 6 (4%), respectively. The number of hips classified into the preoperative NP group was 54 (33%), accounting for approximately one-third of all hips; however, this number decreased significantly to 7 (4%) after THA ($p < 0.01$) (Table 1). Of the seven hips classified into the postoperative NP group, six were also classified into the NP group preoperatively, while one was classified into the non-NP group preoperatively. A significantly higher percentage was observed in the NP group both pre-and postoperatively ($p < 0.01$), indicating that patients with preoperative neuropathic pain tended to have prolonged neuropathic pain (Fig 1). NRS score improved from 8.5 preoperatively to 0.7 points postoperatively in the NP group, and from 6.8 preoperatively to 0.3 points postoperatively in the non-NP group. A significantly higher NRS score was observed in the NP group than in the non-NP group both before and after THA ($p < 0.05$). Furthermore, the mean age was significantly higher in the NP group than in the non-NP group ($p < 0.05$). No significant differences in sex or BMI were observed between the two groups (Table 2); however, a positive correlation was observed between the preoperative PDQ and NRS scores ($r = 0.358$, $p < 0.001$). Similarly, a positive correlation was observed between the preoperative PDQ and postoperative NRS scores ($r = 0.375$, $p < 0.001$) (Fig 2). A correlation was not observed between the preoperative PDQ and BMI ($r = 0.052$, $p = 0.51$).

## Discussion

Osteoarthritic pain has traditionally been considered a nociceptive pain; however, recent studies have reported a close relationship between OA and neuropathic pain [6, 7, 16–19].

**Table 1. Prevalence of the different pain phenotypes in patients with osteoarthritis of the hip.**

|  | Nociceptive Pain | Unclear Pain | Neuropathic Pain | Possible Neuropathic Pain* |
|---|---|---|---|---|
| preoperative | 105(66%) | 36(23%) | 18(11%) | 54(33%) |
| postoperative | 152(95%) | 6(3%) | 1(1%) | 7(4%) |

**Note:** *including neuropathic pain and unclear pain

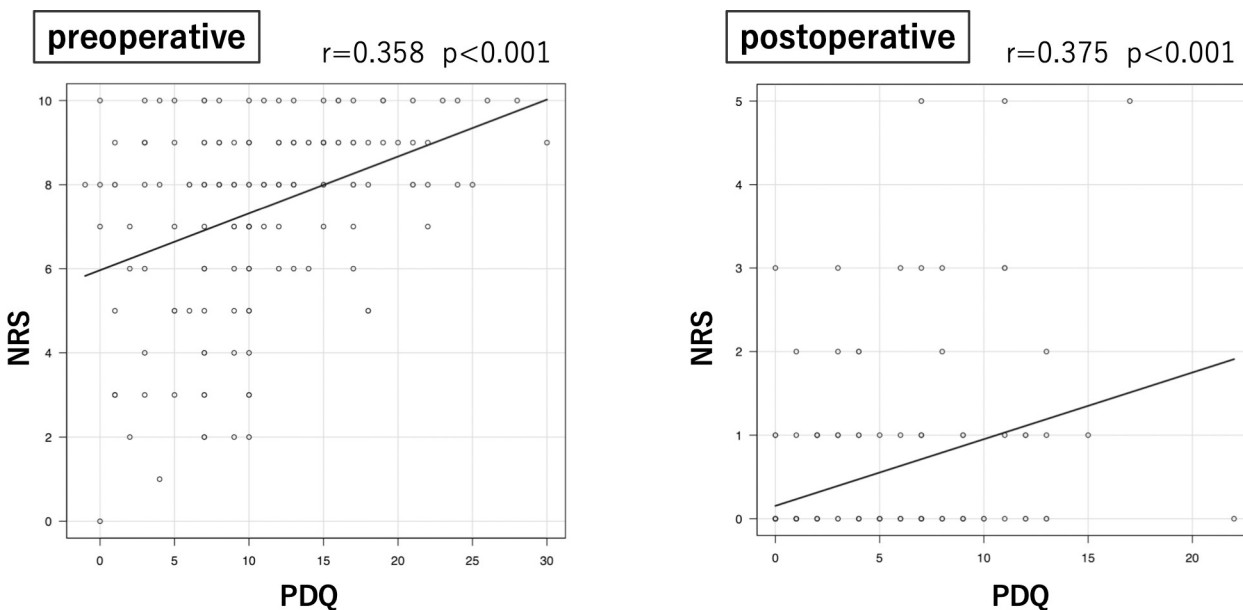

**Fig 1. Breakdown of pain phenotypes before and after total hip arthroplasty (THA).** Abbreviations: NP group, neuropathic pain group; non-NP group, non-neuropathic pain group.

Neuropathic pain is also known to cause chronic postoperative pain in patients with OA. Maeda et al. reported a positive correlation between preoperative PDQ scores before THA and postoperative pain [8], and Hasegawa et al. reported an association between chronic pain after THA and preoperative neuropathic pain [9]. A deeper understanding of the causes of pain in OA has led to the use of centrally acting drugs [10].

Assessment tools such as LANSS [11], S-LANSS [12], NPQ [13], DN4 [14], ID Pain [15], and PainDETECT are widely used to assess neuropathic pain. In Japan, a Japanese version of the PDQ has been developed and is widely used. Using the PDQ, pain can be differentiated with 85% sensitivity and 80% specificity. Despite limited data directly comparing these tools, Mulvey et al. reported that the accuracy of the PDQ is comparable to that of other self-report questionnaires, and its reliability and validity have been confirmed [16]. Bennett et al. reviewed the strengths and limitations of these questionnaires and reported that PainDETECT did not require clinical examination and showed slightly higher sensitivity and specificity than other

**Table 2. Comparison between patients with the preoperative pain phenotypes of possible neuropathic pain (NP group) versus those with nociceptive pain(non-NP group).**

| | | NP group | non-NP group | |
|---|---|---|---|---|
| | | n = 54 | n = 105 | p-value |
| NRS scores | preoperative | 8.54 ± 1.26 | 6.82 ± 2.44 | <0.001 |
| | postoperative | 0.76 ± 1.25 | 0.30 ± 0.77 | <0.01 |
| Gender | male | 10 | 12 | NS |
| | female | 44 | 93 | |
| Age (years) | | 70.17 ± 10.04 | 64.96 ± 9.97 | <0.01 |
| BMI (kg/m$^2$) | | 25.38 ± 4.53 | 24.66 ± 3.76 | NS |

**Note**: Data are shown as mean ± standard deviation

**Abbreviations**: NRS, numerical rating scale; BMI, body mass index; NS, not significant

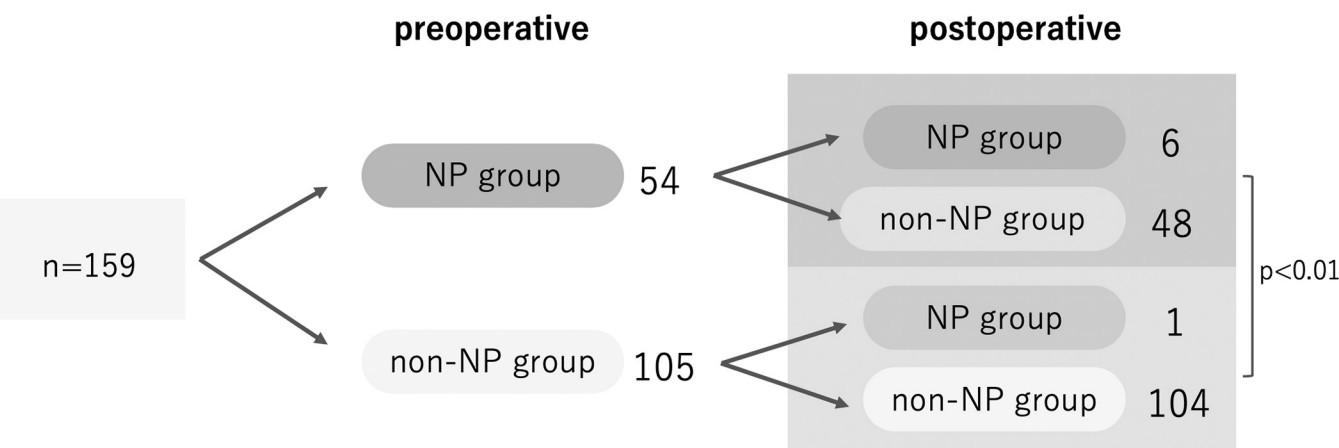

**Fig 2. Correlation between preoperative painDETECT questionnaire (PDQ) score and pre- and postoperative numerical rating scale (NRS).**

tools [17]. However, a challenge with the PDQ is that even a score of 0 is classified as nociceptive pain; thus, even patients without pain are placed into the nociceptive pain group.

In a previous study that evaluated hip OA pain using the PDQ, the prevalence of neuropathic pain with a PDQ score of 13 or higher ranged from 19% to 37% (Table 3) [18–21]. The prevalence of neuropathic pain in the present study was 33%, similar to the frequencies reported in previous studies. However, previous studies have not evaluated preoperative pain outcomes and PDQ scores after THA in patients with neuropathic pain. In this study, we investigated PDQ scores before and after THA and found that THA decreased the prevalence of neuropathic pain. Patients with preoperative neuropathic pain were more likely to experience prolonged postoperative neuropathic pain. Fitzsimmons et al. proposed that in patients with preoperative neuropathic pain, careful preoperative evaluation, including for spinal disease, is necessary to avoid chronic postoperative pain [22]. In this study, all patients underwent an interview and X-ray to detect lumbar spine disease. In addition, a lumbar spine MRI was performed on those patients who seemed to need it. It is difficult to completely rule out the involvement of lumbar spine disease preoperatively and elderly patients with neuropathic pain should be searched more carefully for spinal disease before THA.

A positive correlation between the preoperative PDQ and postoperative NRS scores was observed, suggesting that neuropathic pain may lead to worse outcomes after THA and lower patient satisfaction. Although the causes of neuropathic pain and related factors require further investigation, preoperative evaluation, including PDQ, may be useful in reducing the prevalence of postoperative chronic hip pain.

**Table 3. Prevalence of neuropathic pain in hip osteoarthritis patients in the literature assessed using the painDETECT questionnaire (PDQ).**

|  | Neuropathic pain(%) | Possible neuropathic pain*(%) |
|---|---|---|
| Shigemura et al. [18] | 6 | 19 |
| Rienstra et al. [19] | 0 | 32 |
| Blikman et al. [20] | NA | 37 |
| Power et al. [21] | 10 | 27 |
| Present study | 11 | 33 |

**Note:** *including neuropathic pain and unclear pain

This study has certain limitations, including the shortage of evaluation of spinal disease in patients with hip OA and a short follow-up period of 6 months. A correlation between age and the incidence of neuropathic pain may have occurred because spinal disease could not be completely ruled out. We have used only PDQ and NRS to verify the association between the presence of preoperative neuropathic pain and the occurrence of chronic postoperative pain. The reliability of PDQ for neuropathic pain in patients with hip OA has not been fully evaluated, and further studies are required to clarify this. Surgical approaches have not been unified, and the effect of different approaches on neuropathic pain has not been studied.

## Conclusion

In this study, neuropathic pain was evaluated using the PDQ before and after surgery in patients with hip OA who underwent THA. Possible neuropathic pain was present in 54 of the 159 hips (33%). Patients with preoperative neuropathic pain tended to have higher preoperative and postoperative NRS scores, suggesting that the involvement of neuropathic pain should be considered when treating patients with hip OA. Furthermore, although the prevalence of possible neuropathic pain decreased to 4% after THA, our results indicate an association between the presence of preoperative neuropathic pain and the occurrence of chronic postoperative pain after THA.

## Supporting information

**S1 File.**
(XLSX)

## Acknowledgments

We would like to thank the patients who participated in this study, and Editage (www.editage.com) for English language editing.

## Author Contributions

**Conceptualization:** Yohei Yamabe.

**Writing – original draft:** Yohei Yamabe, Masahiro Hasegawa, Gai Kobayashi, Shine Tone, Yohei Naito, Akihiro Sudo.

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
