## [Decision Letter · Decision Letter 0]

31 Jan 2024

PONE-D-23-44183Neuropathic pain in patients with osteoarthritis of the hip before and after total hip arthroplastyPLOS ONE

Dear Dr. Hasegawa,

Thank you for submitting your manuscript to PLOS ONE. After careful consideration, we feel that it has merit but does not fully meet PLOS ONE’s publication criteria as it currently stands. Therefore, we invite you to submit a revised version of the manuscript that addresses the points raised during the review process.

Thank you for submitting this interesting paper.Below are the comments of the two reviewers who highlighted important doubts about the methodology used and which makes the paper unpublishable in this capacity. 

I hope you can provide a deep revision of the paper so as to make it publishable.

We look forward to receiving your revised manuscript.

Kind regards,

Raffaele Vitiello

Academic Editor

PLOS ONE

Journal Requirements:

2. Thank you for submitting the above manuscript to PLOS ONE. During our internal evaluation of the manuscript, we found significant text overlap between your submission and the following previously published works, some of which you are an author.

10.3349/ymj.2012.53.4.801,

 https://www.dovepress.com/getfile.php?fileID=73952

Therefore, we cannot consider your manuscript as it stands. Please revise the manuscript to rephrase the duplicated text and fully cite all your sources, where appropriate.

3. We note that your Data Availability Statement is currently as follows: “All relevant data are within the manuscript and its Supporting Information files.”

Additional Editor Comments:

Major review according to reviewer

Reviewers' comments:

Reviewer's Responses to Questions

**Comments to the Author**

1. Is the manuscript technically sound, and do the data support the conclusions?

Reviewer #1: Partly

Reviewer #2: Yes

2. Has the statistical analysis been performed appropriately and rigorously? 

Reviewer #1: No

Reviewer #2: Yes

3. Have the authors made all data underlying the findings in their manuscript fully available?

Reviewer #1: Yes

Reviewer #2: Yes

4. Is the manuscript presented in an intelligible fashion and written in standard English?

Reviewer #1: Yes

Reviewer #2: Yes

5. Review Comments to the Author

Reviewer #1: Thank you for giving me an opportunity to review the article.

In this study, the authors investigated the relationships between hip OA and neuropathic pain.

The topic of study was interesting. However, there are some concerns and limitations in this manuscript. I comment as below:

General comments

・In this study, the authors investigated postoperative neuropathic pain using PDQ. In the current results, the number of patients who referred neuropathic pain decreased. From the results, my question was whether the patients were really having neuropathic pain. As the author described, there were some tools to assess the neuropathic pain. I think that the author should perform another assessment.

・Was the timing of the assessment the same for all patients? The author should describe the timing, who assessed, whether NRS on rest or activity, and whether medication such as pain killer was performed or not.

・The patients who had spine disorders should be investigated.

・The surgical approach was described to affect postoperative pain. The surgical approach should be united when the pain study.

・The evidence to prove the association between the presence of preoperative neuropathic pain and the occurrence of chronic postoperative pain after THA was insufficient.

Specific comments

Line 61: I think including “unclear pain” into NP group was not appropriate.

Line 72: As I wrote above, the definition of NP group and non-NP group was not appropriate. The authors should assess the correlations between the score of PDQ and NRS score and BMI.

Line 89: Please show average and standard deviation or standard error on NRS score, Age and BMI.

Reviewer #2: VERY INTERESTING TOPIC.

HOWEVER, I BELIEVE THAT THE LACK OF DATA RELATING TO THE COEXISTENCE OF SPINE DISEASES CAN DETERMINE A BIAS. IN THIS REGARD, HOW DO YOU EXPLAIN THE CORRELATION BETWEEN ADVANCED AGE AND GREATER INCIDENCE OF NEUROPATHIC PAIN? TO OPTIMIZE THE RESULTS IT IS ESSENTIAL THAT AT LEAST IN THIS GROUP OF PATIENTS THE COEXISTENCE OF VERTEBRAL PROBLEMS IS EXCLUDED.

6. PLOS authors have the option to publish the peer review history of their article (what does this mean?). If published, this will include your full peer review and any attached files.

Reviewer #1: No

Reviewer #2: **Yes: **MICHELA SARACCO

---

## [Author Response · Author response to Decision Letter 0]

1 Mar 2024

Responses to the comments by the Associate editor:

[Response]

Thank you for your comment. We have confirmed that our manuscript fits the PLOS ONE’s style.

2. During our internal evaluation of the manuscript, we found significant text overlap between your submission and the following previously published works, some of which you are an author. Please revise the manuscript to rephrase the duplicated text and fully cite all your sources, where appropriate.

[Response]

Thank you for your comment. I have revised the manuscript overlapping based on your suggestion as below. Please let me know if other areas need to be corrected.

(Page 3, Lines 27; Page 4, Lines 69-73; Page 4, Lines 78-100)

3. We note that your Data Availability Statement is currently as follows: “All relevant data are within the manuscript and its Supporting Information files.”Please confirm at this time whether or not your submission contains all raw data required to replicate the results of your study.

[Response]

We have confirmed the above.

[Response]

Thank you for your helpful recommendation. We have moved the ethics statement to the Methods section as below. (Page 4, Lines 71)

“This study was approved by the institutional review board of Mie University Hospital (H2018-083) and all patients received written informed consent to participate.”

Responses to the Comments by the reviewer 1:

・As the author described, there were some tools to assess the neuropathic pain. I think that the author should perform another assessment.

[Response]

Thank you for your valuable suggestions. We understand that we should perform another assessment. The main reason why we chose the PDQ is that we have a Japanese-language version of this screening tool. Because of the language barrier, it isn't easy to use another self-reporting screening tool in our country, especially for elderly patients. In addition, we compared the usefulness of each tool based on the references (5-6,16 in the reference list) and determined that the PDQ is a simple and reliable tool. Unfortunately, it is impossible to attempt another assessment on the same patients from now because of the characteristics of this study. We added the following sentence as a limitation. (Page 9, Lines 176)

“We have used only PDQ and NRS to verify the association between the presence of preoperative neuropathic pain and the occurrence of chronic postoperative pain.”

・Was the timing of the assessment the same for all patients? The author should describe the timing, who assessed, whether NRS on rest or activity, and whether medication such as pain killer was performed or not.

[Response]

Thank you for your great comments. According to your suggestion, we have fixed and added the sentences as below.

(Page 4, Lines 67)

“The PDQ was administered preoperatively and 6 months postoperatively in all patients, and the prevalence of neuropathic pain was measured before and after THA by the surgeon. Sex, age, BMI, and numerical rating scale (NRS) scores on rest were compared between groups.”

(Page 4, Lines 54)

“For the first two weeks after surgery, all patients took oral NSAIDs or acetaminophen according to our protocol. No patients used analgesics specifically for neuropathic pain.”

・The patients who had spine disorders should be investigated.

[Response]

Thank you very much for your excellent comments. We also determined that spine disorders needed to be ruled out, and we examined all patients as much as possible preoperatively. Specifically, all patients underwent an interview and spinal X-ray to detect lumbar spine disease. In addition, a lumbar spine MRI was performed on those patients who seemed to need it. Patients whose hip pain is suspected of neuropathic pain because of spinal disorders were not indicated for THA. According to your suggestion, we have added the following sentences to the Methods section and Discussion section as below.

(Page 4, Lines 52)

“Patients whose preoperative physical and imaging findings suggested that their hip pain was due to spine-related disease were not indicated for THA.”

(Page8, Lines 159)

“In this study, all patients underwent an interview and X-ray to detect lumbar spine disease. In addition, a lumbar spine MRI was performed on those patients who seemed to need it. It is difficult to completely rule out the involvement of lumbar spine disease preoperatively and elderly patients with neuropathic pain should be searched more carefully for spinal disease before THA”

・The surgical approach was described to affect postoperative pain. The surgical approach should be united when the pain study.

[Response]

Thank you for your comment. As you indicated, the surgical approaches should have been unified. Since the various approaches were mixed in the following references (16-19 in the reference list), we did not unite the approaches in this study. We added the following sentence as a limitation. (Page 9, Lines 179)

“Surgical approaches have not been unified, and the effect of different approaches on neuropathic pain has not been studied.”

・The evidence to prove the association between the presence of preoperative neuropathic pain and the occurrence of chronic postoperative pain after THA was insufficient.

[Response]

Thank you for your helpful recommendations. In our research, a significantly higher postoperative NRS score was observed in the preoperative NP group (0.76 ± 1.25) than in the preoperative non-NP group (0.30 ± 0.78) after THA. In addition, a positive correlation was observed between the preoperative PDQ and postoperative NRS scores (r = 0.375, p < 0.001). While there may not be a large difference in the calculated scores, for these reasons we determined that there is an association between the presence of preoperative neuropathic pain and the occurrence of chronic postoperative pain after THA. We added the following sentence as a limitation. (Page 9, Lines 176)

“We have used only PDQ and NRS to verify the association between the presence of preoperative neuropathic pain and the occurrence of chronic postoperative pain.”

Line 61: I think including “unclear pain” into NP group was not appropriate.

Line 72: As I wrote above, the definition of NP group and non-NP group was not appropriate. The authors should assess the correlations between the score of PDQ and NRS score and BMI.

[Response]

Thank you for your comment. We consider this grouping is appropriate because all of the cited references (16-19 in the reference list) that evaluated neuropathic pain using the PDQ set the cutoff value at 13 and included unclear pain into the NP group. Since we have already discussed the correlation between PDQ and NRS, we have added an examination of the correlation between PDQ and BMI to the methods and discussion as below.

(Page 4, Lines 70)

“The correlation between the preoperative PDQ score and the pre- and postoperative NRS scores and between the preoperative PDQ score and BMI were also investigated.”

(Page 4, Lines 80)

“Spearman's correlation coefficient was used to determine the correlations between pre- and post-THA NRS scores and preoperative PDQ scores, and between BMI and preoperative PDQ scores.”

(Page 5, Lines 117)

“A correlation was not observed between the preoperative PDQ and BMI (r = 0.052, p = 0.51).”

Line 89: Please show average and standard deviation or standard error on NRS score, Age and BMI.

[Response]

Thank you for providing these insights. According to your suggestion, we have changed the notation of data as the average and standard deviation on NRS score, age and BMI in Table2 (Page6, Lines 130).

Responses to the Comments by the reviewer 2:

VERY INTERESTING TOPIC.

HOWEVER, I BELIEVE THAT THE LACK OF DATA RELATING TO THE COEXISTENCE OF SPINE DISEASES CAN DETERMINE A BIAS. IN THIS REGARD, HOW DO YOU EXPLAIN THE CORRELATION BETWEEN ADVANCED AGE AND GREATER INCIDENCE OF NEUROPATHIC PAIN? TO OPTIMIZE THE RESULTS IT IS ESSENTIAL THAT AT LEAST IN THIS GROUP OF PATIENTS THE COEXISTENCE OF VERTEBRAL PROBLEMS IS EXCLUDED.

[Response]

Thank you very much for your excellent comments. We also determined that spine disorders needed to be ruled out, and we examined all patients as much as possible preoperatively. Specifically, all patients underwent an interview and spinal X-ray to detect lumbar spine disease. In addition, a lumbar spine MRI was performed on those patients who seemed to need it. Patients whose hip pain is suspected of neuropathic pain because of spinal disorders were not indicated for THA. According to your suggestion, we have added the following sentences to the Methods section and Discussion section as limitation.

(Page 4, Lines 52)

“Patients whose preoperative physical and imaging findings suggested that their hip pain was due to spine-related disease were not indicated for THA.”

(Page8, Lines 159)

“In this study, all patients underwent an interview and X-ray to detect lumbar spine disease. In addition, a lumbar spine MRI was performed on those patients who seemed to need it. It is difficult to completely rule out the involvement of lumbar spine disease preoperatively and elderly patients with neuropathic pain should be searched more carefully for spinal disease before THA”

(Page9, Lines 175)

“A correlation between age and the incidence of neuropathic pain may have occurred because spinal disease could not be completely ruled out.”

---

## [Decision Letter · Decision Letter 1]

15 Mar 2024

Neuropathic pain in patients with osteoarthritis of the hip before and after total hip arthroplasty

PONE-D-23-44183R1

Dear Dr. Hasegawa,

We’re pleased to inform you that your manuscript has been judged scientifically suitable for publication and will be formally accepted for publication once it meets all outstanding technical requirements.

Kind regards,

Raffaele Vitiello

Academic Editor

PLOS ONE

Additional Editor Comments (optional):

Reviewers' comments:

Reviewer's Responses to Questions

**Comments to the Author**

1. If the authors have adequately addressed your comments raised in a previous round of review and you feel that this manuscript is now acceptable for publication, you may indicate that here to bypass the “Comments to the Author” section, enter your conflict of interest statement in the “Confidential to Editor” section, and submit your "Accept" recommendation.

Reviewer #1: All comments have been addressed

2. Is the manuscript technically sound, and do the data support the conclusions?

Reviewer #1: Yes

3. Has the statistical analysis been performed appropriately and rigorously? 

Reviewer #1: Yes

4. Have the authors made all data underlying the findings in their manuscript fully available?

Reviewer #1: Yes

5. Is the manuscript presented in an intelligible fashion and written in standard English?

Reviewer #1: Yes

6. Review Comments to the Author

Reviewer #1: (No Response)

7. PLOS authors have the option to publish the peer review history of their article (what does this mean?). If published, this will include your full peer review and any attached files.

Reviewer #1: No

---

## [Editor Report · Acceptance letter]

20 Mar 2024

PONE-D-23-44183R1 

PLOS ONE

Dear Dr. Hasegawa, 

I'm pleased to inform you that your manuscript has been deemed suitable for publication in PLOS ONE. Congratulations! Your manuscript is now being handed over to our production team.

Kind regards, 

on behalf of

Dr. Raffaele Vitiello 

Academic Editor

PLOS ONE